# PKCθ Regulates Pituitary Adenoma Bone Invasion by Activating Osteoclast in NF-κB/IL-1β-Dependent Manner

**DOI:** 10.3390/cancers15051624

**Published:** 2023-03-06

**Authors:** Quanji Wang, Zhuowei Lei, Zihan Wang, Qian Jiang, Zhuo Zhang, Xiaojin Liu, Biao Xing, Sihan Li, Xiang Guo, Yanchao Liu, Xingbo Li, Kai Shu, Huaqiu Zhang, Yimin Huang, Ting Lei

**Affiliations:** 1Sino-German Neuro-Oncology Molecular Laboratory, Department of Neurosurgery, Tongji Hospital of Tongji Medical College of Huazhong University of Science and Technology, Jiefang Avenue 1095, Wuhan 430030, China; 2Department of Orthopedics, Tongji Hospital of Tongji Medical College of Huazhong University of Science and Technology, Jiefang Avenue 1095, Wuhan 430030, China

**Keywords:** pituitary adenomas, PKCtheta, macrophage, osteoclasts, bone invasion, celastrol

## Abstract

**Simple Summary:**

Pituitary adenoma (PA) bone invasion seriously affects patient prognosis, and its mechanism needs more investigation. Through exploring the interaction between PA tumor cells and osteoclasts, we have revealed an essential theory of the progression of PA bone invasion. Localized inflammatory environment in PAs was identified as a feature of bone invasion. In this study, we identified PKCθ as a key signal for PA bone invasion, and PAs release IL-1β via the PKCθ/NF-κB/IL-1β pathway to induce monocyte–osteoclast differentiation in a paracrine manner. Meanwhile, we also found that celastrol, as a natural product, can obviously reduce the secretion of IL-1β as well as alleviate the progression of bone invasion, which is promising for clinical application.

**Abstract:**

Background: Pituitary adenoma (PA) bone invasion results in adverse outcomes, such as reduced rates of complete surgical resection and biochemical remission as well as increased recurrence rates, though few studies have been conducted. Methods: We collected clinical specimens of PAs for staining and statistical analysis. Evaluation of the ability of PA cells to induce monocyte–osteoclast differentiation by coculturing PA cells with RAW264.7 in vitro. An in vivo model of bone invasion was used to simulate the process of bone erosion and evaluate the effect of different interventions in alleviating bone invasion. Results: We found an overactivation of osteoclasts in bone-invasive PAs and concomitant aggregation of inflammatory factors. Furthermore, activation of PKCθ in PAs was established as a central signaling promoting PA bone invasion through the PKCθ/NF-κB/IL-1β pathway. By inhibiting PKCθ and blocking IL1β, we were able to significantly reverse bone invasion in an in vivo study. Meanwhile, we also found that celastrol, as a natural product, can obviously reduce the secretion of IL-1β as well as alleviate the progression of bone invasion. Conclusions: By activating the PKCθ/NF-κB/IL-1β pathway, pituitary tumors are able to induce monocyte–osteoclast differentiation in a paracrine manner and promote bone invasion, which can be alleviated by celastrol.

## 1. Introduction

Pituitary adenomas (PAs) are benign tumors originating from the anterior pituitary gland with a prevalence of 0.037–0.116% based on clinical case studies, accounting for approximately 10% of all intracranial tumors [1,2]. Although PAs rarely exhibit malignant potential, the abnormal hormone secretion and localized occupying effects can still cause suffering and seriously impact patients’ quality of life. As neuroendocrine tumors, partial PAs are known as functioning PAs with the capacity to secrete hormones including one or more of the following: growth hormone (GH), prolactin (PRL), adrenocorticotropic hormone (ACTH), thyroid stimulating hormone (TSH), gonadotropin (Gn), etc. Approximately 40% of PAs exhibit invasive features [3], and some of the associated factors that were reported include macroadenomas, GH-Pas, and sparsely granulated somatotroph adenomas [4]. Invasive PAs often exhibit characteristics of tissue infiltrative growth that can invade local structures, such as sellar dura, cavernous sinus, brain tissue, and parasellar bone. The incidence of bone-invasive PAs (BIPAs) has been estimated to be approximately 6.4% in total PAs [5], which mainly invade into the sphenoid sinus and clivus. The sphenoid bone serves as a bony barrier adjacent to the pituitary gland, dividing the pituitary gland from the sphenoid sinus. When the sphenoid bone is infiltrated by PAs, it can lead to a poor clinical outcome, including reduced complete surgical resection rates and biochemical remission rates, increased complication rates and recurrence rates, and increased drug resistance [6,7]. 

The mechanisms of bone invasion in PAs are not well understood, and studies have suggested that the invasive characteristics of tumor cells and the appropriate bone microenvironment are essential for bone invasion [8,9]. Previous studies mostly focused on the tumor cells’ enhanced invasive capacity, while the alterations of bone structure warrant more in-depth investigation in the development of bone invasion. The bone surface contains osteoblasts and osteoclasts, cooperating in maintaining the bone homeostasis and stability [10]. Abnormal activation of osteoclasts can lead to the destruction of bone structure and provide a suitable microenvironment for PA bone invasion. Previous studies have reported that invasive PAs can induce osteoclast activation [9,11]. However, the details mechanisms of osteoclast activation in PAs remain largely unknown. Therefore, it is of importance to investigate the progression of PA bone invasion and clarify the participation of osteoclasts in this process to understand and mitigate the bone-invasive behavior of PAs. Zhang et al. reported high local inflammatory response in BIPAs, while TNFα, an inflammatory cytokine, was secreted by PAs cells to activate osteoclasts, hence increasing bone invasiveness [5,11]. The inflammatory status within the tumor and its relationship to osteoclast differentiation require deep exploration.

The PKC family, as serine-threonine kinases, contains 12 members that regulate processes including cell cycle and differentiation, morphogenesis, and cell survival [12]. Previous studies from others and our group revealed that PKC has a central role in promoting hormone secretion [13,14] and tumor proliferation [15,16,17]. PKC mutations are observed in PAs and are considered to be associated with oncogenesis, mainly with PKCα [16,17,18]. PKCθ, which has a similar function to PKCα, had its role in inflammatory activation and a pro-invasive effect in other tumors revealed [19], though studies on PAs are lacking.

In the present study, we aimed to investigate the potential role of PKCθ in bone invasion and its impact on monocyte–osteoclast differentiation. In addition, we also evaluated the treatment efficiency of celastrol on PA bone invasion, which can be translated into clinical therapeutical application. 

## 2. Materials and Methods

### 2.1. Bioinformatics Analysis of PA

Clinical data and RNA sequencing data regarding bone-invasive PA patients were acquired from Zhu et al. [5] and contain 10 samples and 517 differential genes. Gene expression profiles of invasive PAs were downloaded from the Gene Expression Omnibus (GEO; https://www.ncbi.nlm.nih.gov/geo, accessed on 20 January 2023) database. Eligibility criteria included mRNA data from samples of PAs containing both invasive and noninvasive phenotypes. GSE120350, GSE169498, GSE22812, GSE26966, GSE51618, and GSE72490, containing 81 samples of invasive PAs and 55 noninvasive PAs, were included in this study. The datasets were merged and removed batch effects using sva and preprocess Core package before further analysis.

The differentially expressed genes (DEGs) of PAs were screened using the limma package in R 4.2.1. A volcano plot and heatmap are displayed for the DEGs and genes of the PKC family in the GEO expression cohort. Gene set enrichment analysis (GSEA v4.2.3.) of DEGs in invasive Pas through hallmark gene sets and the hallmark gene sets were collected from the Molecular Signatures Database (MsigDB v7.2; https://www.gsea-msigdb.org/gsea/msigdb, accessed on 20 January 2023). Adjusted *p*-value < 0.05 and |normalized enrichment score (NES)| >1 were considered the cutoff values for identifying the enriched pathway. The immune cells infiltration in PAs was analyzed by ssGSEA which is in package GSVA package. We collected cytokines associated with osteoclast differentiation from previous studies [20] as well as GeneCard v5.12 (https://www.genecards.org, accessed on 20 January 2023). 

### 2.2. Clinical Samples

A total of forty human PA specimens were obtained from operations between January 2020 and July 2022 from the Department of Neurosurgery in Tongji Hospital, Wuhan. This study was approved by the Research Ethics Board of Tongji Hospital, and all patients gave informed consent to the study. We classified all patient samples as BIPAs and non-bone-invasive PAs (NBIPAs) based on the bone invasion of the PAs. The criteria for classification were as follows: 1. Preoperative computed tomography (CT) and magnetic resonance imaging (MRI) imaging reports suggesting bone infiltration. 2. Intraoperative recording of pituitary tumor invasion of surrounding bone (mainly sphenoid bone). 3. Postoperative histopathological sections suggesting tumor cell infiltration in bone. In our classification, the first item is used as a reference basis only, with the second item as the main criterion and the data for the third item often lacking. The summary information of the collected patient data is displayed in Table 1.

### 2.3. Animal Model of Bone-Invasive PAs

The animal experiments were conducted in accordance with the experimental protocol approved by the Tongji Hospital Committee for the care of animals (process number: TJH-202206015). Male BALB/c athymic nude mice (Gempharmatech, Ningjing, China) used for molding were housed in an animal facility under SPF conditions. At 6–7 weeks of age, the mice were used to develop an animal model of bone invasion by PAs [21]. GH3, MMQ, TtT/GF (1 × 106 cells/100 µL) in PBS were injected between the galea aponeurotica and pericranium, overlaying the calvaria. All animals were sacrificed at day 15. Surgical acquisition of tumor and skull specimens was performed, and the specimens were fixed in 4% paraformaldehyde for 48 h and were decalcified in 10% EDTA for 15 days. Tumor and skull tissue were embedded in paraffin to prepare 5 μm thick coronal sections for subsequent section staining. All bone invasion specimens were evaluated for bone erosion score based on HE staining. The bone erosion score was calculated using ImageJ v2.1.4.7. and was defined as the defect area in bone compared to the total area of the skull.

To deplete monocytes, beginning at day 2 after injected tumor cells in calvaria, we administered clodronate liposomes (MCE, Shanghai, China) intraperitoneal injection (I.P.) on alternating days with a dosage of 0.05 mg/g bodyweight until euthanasia, while PBS liposomes were utilized as a control. In addition, we adopted different drug administration to alleviate pituitary tumor bone invasion, including sotrastaurin (100 mg/kg, MCE, Shanghai, China), Pyrrolidinedithiocarbamate ammonium (PDTC, 50 mg/kg, MCE, Shanghai, China), IL-1β monoclonal antibodies (B122, 2 mg/kg, Bio X Cell, Lebanon, NH, USA), celastrol (1 mg/kg, MCE, Shanghai, China), and PBS (as control). All drugs were administered via I.P. injection on alternating days beginning on day 7 days after implantation of the tumor cells. 

### 2.4. Tissue Sections Stained

For HE staining, immunohistochemistry (IHC), and immunofluorescence (IF), 5 μm sections were made from clinical specimens and xenograft tumor specimens. For IHC, rehydrated tissue sections were blocked with 5% bovine serum albumin (BSA, ServiceBio, Wuhan, China) and then were stained with primary antibody at 4 ℃ for 12 h. The primary antibodies used include IL-1β (A16288, Abclonal, Wuhan, China), IL-6 (21865-1-AP, Proteintech, Wuhan, China), IL-8 (27095-1-AP, Proteintech, Wuhan, China), TNF-α (GB11188, ServiceBio, Wuhan, China), and PKCθ (13643, Cell Signaling Technology, Danvers, MA, USA). The secondary antibodies used for IHC were biotinylated anti-rabbit IgG (Boster, Wuhan, China). Subsequently, streptavidin–biotin complex (SABC, Boster, Wuhan, China) and DAB (DAB substrate: DAB chromagen = 1 mL: 20 µL) were added to visualize staining, and hematoxylin was used to stain the nucleus. For IF, after rehydration and blocking, samples were incubated with primary antibodies, including F4/80(sc-377009, Santa Cruz Biotechnology, Dallas, TX, USA) and TRAP (ab191406, Abcam, Cambridge, UK), at 4℃ for 12 h. Second antibodies used for IF are Cy3–conjugated Goat Anti-Rabbit IgG and Dylight 488-conjugated Goat Anti-Rabbit IgG (1:200, Proteintech, Wuhan, China), and 4’,6-diamidino-2-phenylindole (DAPI, ServiceBio, Wuhan, China) was used for nuclear staining. All section staining results were observed using microscopy (Olympus, Tokyo, Japan) and analyzed using ImageJ.

### 2.5. Microcomputed Tomography (µCT) Analysis

Tumor and skull specimens obtained from nude mice, after fixation with paraformaldehyde, were scanned using micro-CT (SkyScan 1176, Brüker, Germany) before decalcification processing. Consequently, 3-dimensional images were reconstructed using NRecon software (version 1.7.3). Each calvaria’s area was outlined for quantification, and the amount of bone lost was represented as the proportion of bone volume to tissue volume (BV/TV).

### 2.6. Cells Treatment and Transfection

A total of five cell lines were used in this experiment: pituitary tumor cell lines (GH3, ATT20, and TtT/GF), macrophage cell line (RAW264.7), and 293T, which were obtained from American Type Culture Collection (ATCC, Manassas, VA, USA). All five cell lines were cultured using DMEM/High Glucose medium (ServiceBio, Wuhan, China) supplemented with 10% fetal bovine serum (TransSerum FQ Fetal Bovine Serum, Beijing, China) and 1% antibiotics (ServiceBio, Wuhan, China). Drugs treated in this study included PMA (100 nM), Sotrastaurin (5 nM), RANKL (30 ng/mL), PDTC (30 µM), IL-1β mAb (100 pM), and celastrol (5 μM).

We used short hairpin RNAs (shRNAs) in PA cell lines (GH3, ATT20, TtT/GF) to knock down PKCθ. shRNAs were designed through RnaiDesigner (https://rnaidesigner.thermofisher.com, accessed on 20 January 2023) based on the sequence for mouse PKCθ mRNA (NM_008859) and rat PKCθ mRNA (NM_001276721.1). For the mouse PKCθ target sequence, sh1, GCAAGAATGTAGACCTCATCT; sh2, GCAAGATACTTTCTGGAAATG; sh3, GCAAGATACTTTCTGGAAATG. For the rat PKCθ target sequence, sh1, GCAAGATACTTTCTGGAAATG; sh2, GGACCAATTGAAATCAGTTTC; sh3, GCAACTTCTCCTTCATTAACC. After ligating the shRNA to the plko.1 plasmid, the target plasmid and capsids plasmid were transfected into the 293T cell line using the lipo3000 transfection reagent (Gibco, Thermo Fisher, Waltham, MA, USA). The viral was collected from the 293T medium after 72 h and used for the culture of target cells, and polybrene was given. Green fluorescence can be observed in fluorescence microscopy after successful shRNA transfection, and WB and qPCR were performed to verify knockdown efficiency.

### 2.7. Cell Viability Assays

Cells were seeded in 96-well plates at a density of 5 × 10^3^ cells/well. After 24 h of incubation in a CO_2_ incubator, the medium was replaced with DMEM medium containing the drug. After a certain time of drug action, Cell Counting Kit 8 (Servicebio, Wuhan, China) was added, and the optical density (OD) value at 450 nm was measured after 1 h in the incubator at 37 °C.

### 2.8. Ki-67

The cells were seeded in 24-well plates, and when the cell density reached 70–80%, they were fixed with 4% paraformaldehyde. Cells were blocked for 1 h by 5% BSA within 0.2% Triton X-100 and were incubated with antibodies against Ki-67(ab15580, Abcam, Cambridge, UK) overnight in 4 °C. After incubation with Cy3–conjugated goat anti-rabbit IgG and DAPI in the dark for 2 h, they were photographed under the microscope. Calculation of the proportion of Ki-67 positive cells using ImageJ used a suitable threshold value.

### 2.9. Wound Healing Assay

Cells were seeded in 6-well plates, and when the cell density reached 100%, a straight line was drawn vertically using a 200 μL tip. Cells were washed with PBS, and the DMEM medium containing the drug was replaced and photographed using a microscope as the state of the cells at 0 h. The cells were placed in a CO_2_ incubator and photographed at 12 h, 24 h, 36 h, and 48 h, respectively. The widths of the intercellular gaps were detected using ImageJ, and the wound width at 48 h versus 0 h was used as an indicator of cell migration ability.

### 2.10. Transwell Assay

Matrigel matrix gel (Corning, New York, NY, USA) pre-placed at 4 °C was 1:8 diluted by PBS; 100 μL was added to the upper Transwell chamber (Corning, New York, NY, USA) and then stood for 3 h at room temperature. 10^4^ cells/chamber were seeded in the upper Transwell chamber after resuspension using serum-free DMEM medium. DMEM medium containing 10% FBS was added to the lower Transwell chambers. The Transwell chambers were incubated in a CO_2_ incubator for 24 h, fixed with 4% paraformaldehyde for 30 min, and then stained with crystal violet for 15 min. Cells in the upper chamber were carefully wiped off using a cotton swab, followed by microscopic photography. Cells migrating to the lower chamber were identified using ImageJ, and their count represents the invasion ability of the cells.

### 2.11. Osteoclast Differentiation Induction

RAW264.7 was used to induce osteoclasts. After implantation of RAW264.7 5 × 10^3^ cells/well into 96-well plates, pituitary-tumor-conditioned medium or drug treatment was given accompanied by a lower level of RANKL (30 ng/mL). The medium was changed every 2 days, and the end point of induction was reached after 6 days of culture. After fixation with 4% paraformaldehyde for 30 min, a TRAP staining kit (Servicebio, Wuhan, China) was used to evaluate the effect of osteoclast differentiation. TRAP-positive multinucleated cells were considered to differentiate into osteoclasts and were counted under a microscope.

### 2.12. Quantitative PCR (qPCR)

Total RNA was extracted from PBS-washed cells using TRIzol (50 mM Tris-HCl (pH 7.4), 150 mM NaCl, 1 mM EDTA-2Na, 1% Triton X-100, 1% Sodium deoxycholate and 0.1% SDS, Servicebio, Wuhan, China). Subsequently, 0.5 μg of RNA was reverse transcribed into cDNA using the Hifair* III 1st Strand cDNA Synthesis Kit (Yeasen, Shanghai, China) according to the instructions of the manufacturer. The target nucleic acid fragment in cDNA was amplified and detected on the ABI-Prism 7500 Real-Time PCR System (Applied Biosystems, Carlsbad, CA, USA) using Hieff* qPCR SYBR Green Master Mix (Low Rox Plus) (Yeasen, Shanghai, China). The content of GAPDH was used as an internal reference to correct the target mRNA. The primers used are shown in Appendix A.

### 2.13. Western Blotting

Cellular protein was extracted using RIPA Lysis Buffer (25 mM Tris-HCl, 150 mM NaCl, 1 mM EDTA and 1% NP-40, Servicebio, Wuhan, China) and protease inhibitor cocktail, and the protein concentration was detected using BCA (Servicebio, Wuhan, China). Loading buffer (Servicebio, Wuhan, China) was added before boiling for 10 min for use. For electrophoresis, 10% SDS-PAGE gel was used transferred to a polyvinylidene difluoride (PVDF) membrane, and blocked with 5% skimmed milk powder. Primary antibodies used for incubation were PKCθ (13643, Cell Signaling Technology, Danvers, MA, USA), IKKα (2682, Cell Signaling Technology, Danvers, MA, USA), p-IKKα/β (2697, Cell Signaling Technology, Danvers, MA, USA), IKBα (10268-1-AP, Proteintech, Wuhan, China), p-IKBα (2859, Cell Signaling Technology, Danvers, MA, USA), P65 (ab7970, Abcam, Cambridge, UK), p-P65 (3033, Cell Signaling Technology, Danvers, MA, USA), IL-1β (A16288, Abclonal, Wuhan, China), and β-Tubulin (10094-1-AP, Proteintech, Wuhan, China). After washing off the primary antibody with PBST, the HRP-conjugated anti-rabbit or mouse antibody (Proteintech, Wuhan, China) was incubated for 2 h at room temperature. After adding ECL Chemiluminescent kit (NCM Biotech, Suzhou, China), the target proteins were visualized using the GeneGnome XRQ system (Syngene, Cambridge, UK).

### 2.14. Coimmunoprecipitation (co-IP)

After washing with PBS, cellular proteins were extracted using IP lysis buffer (Servicebio, Wuhan, China) and a phosphorylated protease inhibitors cocktail. The protein samples were incubated with PKCθ antibodies bound to Protein Magnetic Beads (MCE, Shanghai, China), then use a magnet to screen out proteins that bind to PKCθ and eluted using SDS-PAGE Loading Buffer. Western blotting was performed on the the IP samples using antibodies against IKKα, p-IKKα/β, IKBα, and p-IKBα, respectively.

### 2.15. Enzyme-Linked Immunosorbent Assay (ELISA)

Collection of pituitary tumor cell supernatant and detection of IL-1β in cell supernatants using mouse or rat IL-1β ELISA Kits (Elabscience, Wuhan, China) were performed according to the manufacturer’s instructions. The ELISA assay is based on an antibody sandwich method using microtiter plates with coated antibodies on surfaces. We added 100μL of sample to each well and incubated at 37 °C for 90 min. Biotinylated anti-rat IL-1β antibody and horseradish-peroxidase-labeled avidin were added sequentially. We added tetramethylbenzidine (TMB) and measured the OD value with a microplate reader at a wavelength of 450 nm. The IL-1β concentration in the samples was calculated by drawing a standard curve and was normalized to the cell content of each group.

### 2.16. Statistical Analysis

The GraphPad Prism (version 9.0) and R version (4.2.0) were used for statistical analyses. All data for this study will be presented as mean ± standard deviation. One-way ANOVA was used for multisample comparison, and Dunnett’s *t*-test was used for comparison between two groups.

## 3. Results

### 3.1. Bone Invasion of PA Is Associated with Proinflammatory Signals

To investigate potential regulative signals that affect PA bone invasion, in silico analysis was performed on previously published datasets [5]. By comparing the transcriptomic data from PA with or without bone invasion, we observed the notable enrichment of inflammation-related pathways in PAs with bone invasion based on gene set enrichment analysis (GSEA), GO, and KEGG analysis (Figure 1A–C), indicating the participation of inflammation in bone invasion. To further verify this, we detected the inflammation-related signatures in PA samples collected from patients. As shown in the heatmap in Figure 1D, PAs with bone invasion display higher levels of proinflammatory signatures. Moreover, the protein levels of proinflammatory factors, including TNFα, IL-1β, IL-6, and IL-8, were also determined in the PA samples. As expected, PA samples with bone invasion express more proinflammatory factors (Figure 1E). Additionally, to further verify this, we established a PA bone invasion model (Figure 1F). Interestingly, tumor cells expressing proinflammatory factors aggregate more in the bone-adjacent area compared to tumor core or peripheral places (Figure 1G). Taken together, our results illustrate that proinflammatory signatures are related to PA bone invasion.

### 3.2. PKCθ Is Potentially the Key Factor Related to Inflammation Status of PAs

To understand what signal modulates PA inflammation phenotype, we performed comprehensive in silico meta-analysis based on 6 independent PA transcriptome datasets with a total of 136 patients (Figure 2A). The normalization was conducted to remove the batch effect among different sequencing datasets (Appendix A), and the principal component analysis (PCA) figure (Figure 2B) indicates acceptable normalization. 

Next, these 136 RNA-seq data were divided into “inflammation high” and “inflammation low” groups according to expression level of inflammation signature, using the median level as a cutoff (Figure 2C,D). We then compared the different expressed genes between “inflammation high” and “inflammation low” groups, and the GSEA was used to identify regulative signaling pathway. These different expressed genes were highly enriched in the NF-κB signaling pathway (Figure 2E), indicating that NF-κB signaling potentially regulatively signals the PA inflammation status. We further plotted the signatures enriched in the NF-κB pathway by their expression and revealed that PRKCQ (PKCθ) is the most upregulated signature (Figure 2F, Appendix A). PKCθ was revealed as an important player in promoting the inflammatory process [22], but its regulative function remains unknown in PA, especially in PA bone invasion. The expression level of PKCθ was determined in our cohort of patients with PA. We observed notably higher levels of PKCθ as well as occurrence of bone invasion in relatively more aggressive PA evaluated using Knosp classification, tumor volume, Ki-67 staining (Figure 2G, Table 1).

### 3.3. PKCθ Is Involved in the Proliferation, Migration, and Bone Invasion of Pituitary Tumors

To investigate the role of PKCθ in invasive Pas, we first constructed PKCθ knock-down in GH3, ATT20, and TtT/GF cell lines (Appendix A) in combination with reagents targeting PKCθ to evaluate the effect of PKCθ on the malignant progression of PAs. Consistent with previous reports, activation of PKC by PMA resulted in a significant increase in tumor cell activity (CCK8, Appendix A), proliferation (clone formation, Figure 3A; Ki-67, Figure 3C), migration (wound healing assay, Figure 3E), and invasion (Transwell, Figure 3G). The reverse trend was observed following the inhibition of PKCθ by sotrastaurin (a pivotal PKCθ inhibitor; CCK8, Appendix A; clone formation, Figure 3A; Ki-67, Figure 3C and Appendix A; wound healing assay, Figure 3E; Transwell, Figure 3G) or PKCθ knockdown (CCK8, Appendix A; clone formation, Appendix A and Figure 3B; Ki-67, Appendix A, and Figure 3D; wound healing assay, Appendix A and Figure 3F; Transwell, Appendix A and Figure 3H). These results suggest that PKCθ, like other members of the PKC family, also exerts significant pro-tumor effects in pituitary tumors. We next implanted tumor cells into nude mice above the calvaria to construct a PA bone invasion xenograft model in vivo. When implanted with PKCθ knock-down PA cells, consistent with in vitro experiments, a significant reduction in tumor size could be found (Figure 3I,J and Appendix A). Interestingly, by scanning HE-stained tumor and bone specimen sections, significant bone defects could be found in the bone areas adjacent to the tumor (Figure 3K), which were used to evaluate the degree of bone erosion. We found that tumors with PKCθ knockdown had significantly reduced bone erosion compared to negative controls (Figure 3L and Appendix A). Altogether, we confirmed the multifaceted pro-tumor effects of PKCθ in pituitary tumors, which strikingly promoting tumor bone invasion.

### 3.4. PKCθ of Pituitary Adenoma Affects Tumor Bone Invasion by Regulating Monocyte-osteoclast Differentiation

To understand the potential regulative mechanism of PKCθ in PA bone invasion, we first divided the integrated RNA-seq dataset (described in Figure 2A) into PKCθ highly and lowly expressed groups. The profiles of tumor-infiltrated immune cells were assessed using ssGSEA, and macrophage population was predicted to significantly increase in the PKCθ high group, indicating PKCθ expression level with macrophage density (Figure 4A). Indeed, osteoclast or its originated monocyte was a crucial participator in tumor bone invasion in various cancers. To testify this in PA, we generated an in vivo bone invasion model (described in Figure 1F,G), and osteoclast was detected using TRAP. As shown in Figure 4B,C, bone tissue exhibits bone erosion adjacent to tumor cells, while bone erosion also occurs where tumor cells are not infiltrated, suggesting that “releasing factors” might participate in the bone erosion. To further verify the crucial role of PKCθ in PAs as well as monocyte in bone invasion, monocytes were depleted by clodronate liposome in the mice with inoculation of PKCθ knockdown PA (Figure 4D,E,I and Appendix A). Interestingly, we observed that PKCθ knockdown on PA cells resulted in notably decreased levels of bone invasion (Figure 4F,G and Appendix A) and tumor growth (Figure 4H). 

In addition, as indicated by Figure 3B, we believe that “releasing factors” play important roles in bone erosion. Therefore, conditioned medium (CM) from primary PA cells or PKCθ knockdown PA cells after PKC stimulation was collected to treat monocyte (Figure 4J). Remarkably higher levels of osteoclast differentiation were observed on monocyte after stimulation with CM from primary PA cells (Figure 4K). Moreover, PKCθ knockdown on PA cells significantly attenuated monocyte–osteoclast differentiation compared to negative controls (Figure 4L and Appendix A). Sotrastaurin was also added to the PA cells, and CM was harvested to stimulate monocyte. PKCθ inhibition significantly alleviated CM from PA-cell-induced monocyte–osteoclast differentiation (Figure 4M and Appendix A). Taken together, our results illustrated that PA PKCθ mediated monocyte–osteoclast differentiation by releasing soluble factors.

### 3.5. PKCθ of Pituitary Adenoma Promotes Monocyte–Osteoclast Differentiation by Regulating IL-1β Expression and Release

To identify which “releasing factors” released by PAs that mediate monocyte-osteoclast differentiation, we first analyzed the DEGs of osteoclastogenic cytokines between the PKCθ-high and low groups in silico. The present results showed that IL-1β was the only osteoclastogenic cytokine with statistically significant upregulation in the PKCθ-high expression group (Figure 5A). Moreover, IL-1β was also expressed upregulated in invasive PAs (Appendix A). We also verified in our clinical specimens that the levels of IL-1β in PAs were highly correlated with tumor bone invasion (Figure 5B) and PKCθ levels (Figure 5C). We next tested IL-1β release levels in PAs cell using ELISA, which showed that IL-1β release levels were raised by activated PKC and were significantly decreased by PKCθ knockdown or after using PKCθ inhibitors (Figure 5D,E), suggesting that IL-1β secretion levels are regulated by intracellular PKCθ activity. We investigated the involvement of released IL-1β in the development of bone invasion. Administration of recombinant protein of IL-1β enhanced the ability of CM of PA cells to induce monocyte-osteoclast differentiation (Figure 5F,G and Appendix A) also enhanced the bone invasion of PAs in vivo (Figure 5H and Appendix A). This indicates that IL-1β is indeed able to enhance the effect of bone invasion by promoting monocyte–osteoclast differentiation, even in the PKCθ knockdown PAs. Next, to further substantiate that PKCθ induces monocyte–osteoclast differentiation by regulating the release of IL-1β, we administered the IL-1β monoclonal antibody (IL-1β mAb). The ability of CM to promote monocyte–osteoclast differentiation was significantly reversed after the blockade of IL-1β (Figure 5F,I and Appendix A). Similarly, IL-1β mAb attenuated the bone erosion effect of PKC-activated PAs in vivo (Figure 5J and Appendix A). This indicates that activated PKCθ promotes monocyte–osteoclast differentiation and contributes to the progression of bone invasion mainly through the secretion of IL-1β.

To explore the regulatory pathway of PKCθ on IL-1β, we focused on the NF-κB pathway, which is enriched in invasive PAs (described in Figure 2E). Meanwhile, NF-κB, as an inflammatory regulatory pathway regulating the expression of IL-1β, was reported to be affected by the activity of PKCθ [23,24]. We conducted western blot within GH3 and ATT20, exhibiting that the NF-κB pathway was significantly upregulated after the activation of PKC by PMA and was inhibited after PKCθ knockdown or administration of sotrastaurin (Figure 5K–N). This suggests that NF-κB pathway activity can be regulated by PKCθ in PA cell lines. We further investigated whether PKCθ directly regulates NF-κB signaling and to clarify the site of action. Analysis from the STRING website implied that PKCθ could be able to interact with IKK or IKB proteins (Figure 5O). We next used coimmunoprecipitation to obtain proteins interacting with PKCθ in PA cell lines, and western blot showed that activated PKCθ could phosphorylate IKK (Figure 5P). In conclusion, we identified the regulatory network that PKCθ upregulates IL-1β expression through NF-κB signaling (Figure 5Q).

### 3.6. Celastrol Alleviates the Bone Invasion of PAs

As a natural compound, celastrol has been mainly focused on its anti-inflammatory [25] and immunosuppressive effects [26] as well as its antitumor effects [27]. Previous studies have revealed that celastrol can inhibit the RANK pathway in osteoclasts and thus inhibit osteoclast differentiation, making it a therapeutic prospect in diseases related to bone destruction. [28]. In addition, celastrol has also been shown to inhibit multiple inflammatory factors, including IL-1β, by targeting the NF-κB pathway [25]. Accordingly, we hypothesize that celastrol might work as a multiple-site therapeutic agent for bone invasion of PAs. We evaluated the effect of celastrol in inhibiting IL-1β release from PA cells, which was superior to PDTC (NF-κB inhibitor) and sotrastaurin (Figure 6A). Moreover, the ability of PA cells to promote monocyte–osteoclast differentiation was significantly attenuated at the administration of –elastrol (Figure 6B and Appendix A). We then used the bone invasion in vivo model to evaluate the effect of celastrol in alleviating bone invasion. At the administration of celastrol, there was a significant reduction in tumor size (Figure 6C). Importantly, significant remission of bone erosion was observed with Micro-CT and HE staining (Figure 6D–F). Taken together, a considerable effect of celastrol in alleviating bone invasion in Pas has appeared, which provides experimental evidence for further clinical application.

## 4. Discussion

Tumor cell invasion is a complex process of penetration of tissue barriers that involves various of processes, such as epithelial–mesenchymal transition, degradation of the extracellular matrix components, and migration. In the present work, we constructed a in vivo model of bone invasion and investigated the alteration of tumor microenvironment in PAs during bone invasion by which overactivation of osteoclasts was observed. Osteoclasts are involved in the progression of PA bone invasion, eroding the bone adjacent to the tumor and providing available space for tumor invasion. We further demonstrated that the inflammatory factor secretion of PAs contributes to the transformation of monocytes into osteoclasts. As in rheumatoid arthritis and periodontal disease, the local inflammatory environment led to structural remodeling of the bone mass with reduced trabecular bone by activating osteoclasts [29,30], which is consistent with what we have observed in PA bone invasion. By culturing monocytes in the CM of primary PAs, we found that PAs can release substances that induce the conversion of monocytes into osteoclasts. IL-1β as an osteoclastogenic cytokine with the most significantly increased expression in invasive PAs, suggesting that it may participate in developing bone invasion in PAs. It was proved that by inhibiting the release and action of IL-1β, the progression of bone invasion could be effectively alleviated.

Furthermore, we reveal the central role of PKCθ in BIPAs. It is well established that activation of the PKC pathway is involved in the initiation and progression of PAs [13,17]. We found that PKCθ, the most significantly increased expression member of the PKC family in silico, was closely associated to PA proliferation and inflammatory pathway activation in invasive PAs. In this experiment, we employed PKCθ inhibitors and PKCθ knockdown to demonstrate that PKCθ is involved in the proliferation and invasion of PAs and related to the monocyte–osteoclast differentiation. We used coimmunoprecipitation to prove that in PAs, PKCθ can phosphorylate IKK, promote the transcriptional activity of P65, and thus promote the expression of IL-1β [31]. Therefore, we established that PKCθ transcriptionally upregulates the expression and release of IL-1β by activating the NF-κB pathway.

Regarding the relationship between different hormone types of PAs and bone invasion, we have yet to find any strong correlation in our experiments. In previous clinical case analysis, it was believed that GH-PAs had a higher proportion of bone invasion. In addition, patients with ACTH-PAs are more prone to osteoporosis and presumably may promote bone invasion [32]. We used two cell lines, GH3 and ATT20, to conduct in vivo experiments, and the bone invasions of the two cell lines were similar, which may be due to the limitation of the bone invasion model.

Finally, we showed the superiority of celastrol in alleviating bone invasion, with clinical application prospects, when compared to the other three drugs (PDTC, sotrastaurin, and IL-1β mAb). Studies have reported that celastrol, as a multisite inhibitor of osteoclast differentiation, can inhibit the NF-κB/IL-1β pathway as well as the activation of the RANK pathway in monocytes [25,33]. Additionally, celastrol is also effective in inhibiting tumor progression and is expected to be an adjuvant drug for the treatment of PAs, especially bone-invasive PAs. In patients with BIPAs, the rational use of drugs to alleviate bone invasion may provide an opportunity to avoid surgery. In addition, postoperative administration of drugs for controlling bone invasion could be an effective method of reducing recurrence.

## 5. Conclusions

PA bone invasion results in adverse outcomes such as reduced rates of complete surgical resection and biochemical remission, and increased recurrence rates. Through exploring the interaction between PA tumor cells and osteoclasts, we have shown that the release of inflammatory factors, mainly IL-1β, of PAs drives monocyte–osteoclast differentiation and aggravates the progression of bone invasion. Furthermore, activation of PKCθ in PAs was established as a central signaling promoting PA bone invasion through the PKCθ/NF-κB/IL-1β pathway. We also found that celastrol, as a natural product, can obviously reduce the secretion of IL-1β as well as alleviate the progression of bone invasion, which is promising for clinical application.

## Figures and Tables

**Figure 1 cancers-15-01624-f001:**
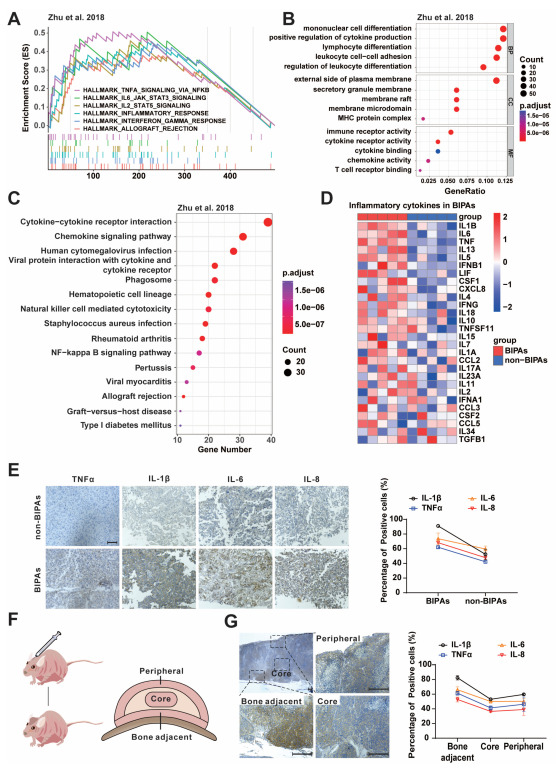
Inflammatory factors in BIPAs. (**A**–**C**) Multi-GSEA analysis (**A**), GO analysis (**B**), and KEGG (**C**) analysis of PA transcriptomic data in the BIPAs versus the non-BIPAs (RNAseq data from the article Zhu et al., 2018 [5]). (**D**) qPCR of major inflammatory cytokines between BIPAs and non-BIPAs. (**E**) Immunohistochemistry for TNFα, IL-1β, IL-6, and IL-8 in clinical specimens of GH-PAs between BIPAs and non-BIPAs (×200; scale bar, 100 μm). Quantification, right. (**F**) Schematic diagram of calvaria xenograft PAs in nude mice. The tumors were specimenized with the skull on the coronal plane; the tumor regions were divided into 3 areas as the peripheral area, core area, and bone-adjacent area. (**G**) The distribution of IL-1β in the xenograft tumor by immunohistochemistry (top left, ×40; scale bar, 200 μm). The localized images of different areas are magnified separately for the peripheral area (top right, ×200; scale bar, 200 μm), core area (bottom right, ×200; scale bar, 200 μm), and bone-adjacent area (bottom left, ×200; scale bar, 200 μm). Quantification of TNFα, IL-1β, IL-6, and IL-8, right.

**Figure 2 cancers-15-01624-f002:**
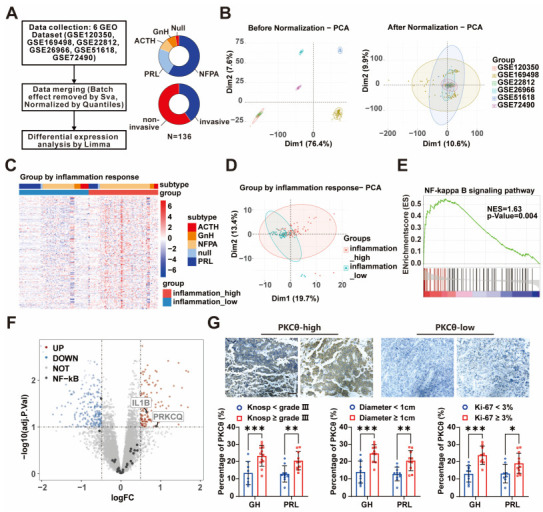
PKCθ and inflammatory response pathways in PAs. (**A**) flowchart showing PA transcriptome data collected from GEO (GSE120350, GSE169498, GSE22812, GSE26966, GSE51618, and GSE72490) merged for differential analysis; the phenotype classification statistics are shown to the right. (**B**) Scatter plots show the principal component analysis (PCA) of normalized gene expression data before (left) and after (right) batch effect removal. (**C**,**D**) Heatmap (**C**) and PCA (**D**) plot shows PAs samples divided into high inflammatory response group and low inflammatory response group. (**E**) GSEA analysis showed that the upregulated genes in the high inflammatory response group were mainly enriched in the NF-κB pathway (NES = 1.63, *p* = 0.004). (**F**) Volcano plot of DEGs between high inflammatory response group and low inflammatory response group, emphasizing for core enriched genes in NF-κB pathway. (**G**) Immunohistochemistry for PKCθ in clinical specimens of GH-PAs (*n* = 20) and PRL-PAs (*n* = 20) (×200; scale bar, 100 μm). Quantification within tumor diameter, Knosp grade, and Ki-67 levels (*** *p* < 0.001; ** *p* < 0.01; * *p* < 0.05).

**Figure 3 cancers-15-01624-f003:**
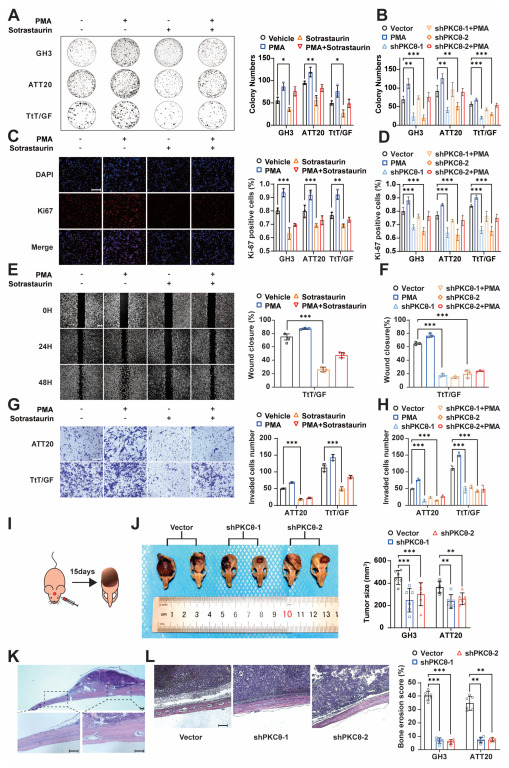
PKCθ is associated with proliferation, migration, and bone invasion of PAs. (**A**,**B**) The colony formation of PA cell lines treated with PMA and sotrastaurin (**A**). Quantification, right. The statistical chart of colony formation of shPKCθ in PA cell lines (**B**). (*n* = 3 independent experiments; *** *p* < 0.001; ** *p* < 0.01; * *p* < 0.05) (**C**,**D**) The Ki-67 staining of GH3 cells treated with PMA and sotrastaurin (**C**) (×200; scale bar, 200 μm). Quantification with ATT20 and TtT/GF, right. The statistical chart of Ki-67 staining of shPKCθ in PA cell lines (**D**). (*n* = 3 independent experiments; *** *p* < 0.001; ** *p* < 0.01) (**E**,**F**) The wound healing assay of TtT/GF cells treated with PMA and sotrastaurin (**E**) (×100; scale bar, 200 μm). Quantification, right. The statistical chart of wound healing assay of shPKCθ in TtT/GF cells (**F**) (*n* = 3 independent experiments; *** *p* < 0.001). (**G**,**H**) The Transwell assay of ATT20 and TtT/GF cells treated with PMA and sotrastaurin (**G**) (×200; scale bar, 200 μm). Quantification, right. The statistical chart of wound healing assay of shPKCθ in ATT20 and TtT/GF cells (**H**) (*** *p* < 0.001). (**I**) Schematic diagram of calvaria xenograft PA in nude mice. (**J**) Photographs of GH3 calvaria xenograft tumor specimens in groups of vector and shPKCθ. Quantification of tumor size, right. (*n* = 6 mice per group; *** *p* < 0.001; ** *p* < 0.01) (**K**) HE staining of tumor coronal section (top; ×40; scale bar, 200 μm), local magnification (bottom; ×100; scale bar, 200 μm). (**L**) HE staining exhibited bone destruction of GH3 tumors in groups of vector and shPKCθ (×40; scale bar, 500 μm). Quantification of bone erosion score, right. (*n* = 6 mice per group; *** *p* < 0.001; ** *p* < 0.01).

**Figure 4 cancers-15-01624-f004:**
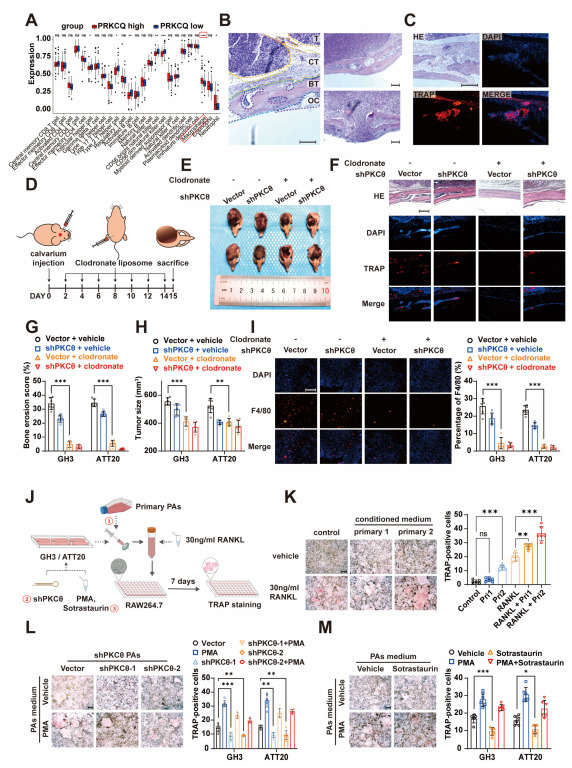
PAs induce monocyte–osteoclast differentiation via the paracrine pathway. (**A**) Immune infiltration analysis; PRKCQ high expression group versus PRKCQ low expression group (the data analyzed are described in Figure 2A; *** *p* < 0.001; ** *p* < 0.01; * *p* < 0.05; NS *p* ≥ 0.05). (**B**) HE staining exhibited the local environment of tumor–bone junction area (×100; scale bar, 200 μm; T, tumor; CT, connective tissue; BT, bone tissue; OC, osteoclast). (**C**) Immunofluorescence of TRAP+ cells in the tumor–bone junction area (×200; scale bar, 100 μm). (**D**) Schematic diagram of calvaria xenograft PA in nude mice and clodronate liposome administration. (**E**,**H**) Photographs of GH3 calvaria xenograft tumor specimens in treatment of clodronate liposomes and shPKCθ (**E**). Quantification of tumor size (**H**) (*n* = 6 mice per group; *** *p* < 0.001; ** *p* < 0.01). (**F**,**G**) HE staining and immunofluorescence of TRAP+ cells in the tumor–bone junction area (**F**) (×200; scale bar, 200 μm). Quantification of bone erosion score (**G**) (*n* = 6 mice per group; *** *p* < 0.001). (**I**) Immunofluorescence of F4/80+ cells in the xenograft tumor (×200; scale bar, 200 μm). Quantification, right (*n* = 6 mice per group; *** *p* < 0.001). (**J**) Schematic diagram of PAs; CM and 30 ng/mL RANKL were administered to induce RAW264.7 differentiation towards osteoblasts. (**K**) TRAP staining of RAW264.7 administered with primary PAs CM (Pri1, primary PAs 1, non-BIPA); (Pri2, primary PAs 2, BIPA); and 30 ng/mL RANKL (×40; scale bar, 500 μm). Quantification, right (*n* = 6 independent experiments; *** *p* < 0.001; ** *p* < 0.01; NS *p* ≥ 0.05). (**L**) TRAP staining of RAW264.7 administrated of 30 ng/mL RANKL and CM of GH3 and ATT20 cells in groups of vector and shPKCθ (×40; scale bar, 500 μm). Quantification, right (*n* = 6 independent experiments; *** *p* < 0.001; ** *p* < 0.01; * *p* < 0.05). (**M**) TRAP staining of RAW264.7 administrated of 30 ng/mL RANKL and CM of GH3 and ATT20 cells in treatment of sotrastaurin and PMA (×40; scale bar, 500 μm). Quantification, right (*n* = 6 independent experiments; *** *p* < 0.001; ** *p* < 0.01; * *p* < 0.05).

**Figure 5 cancers-15-01624-f005:**
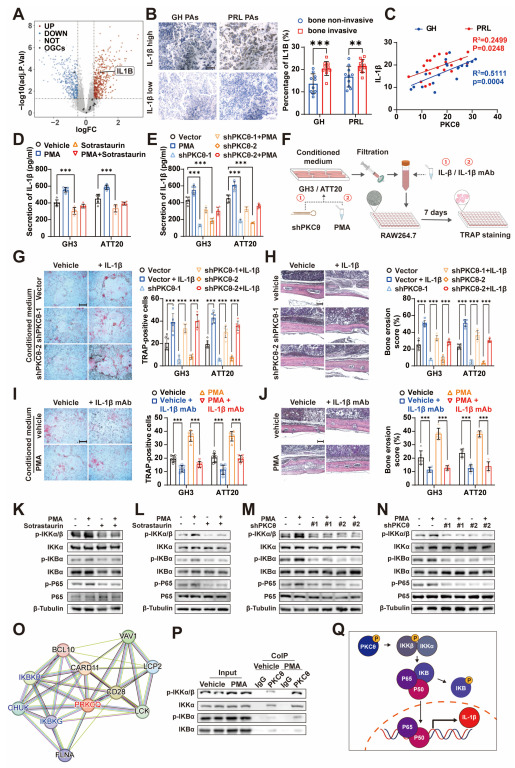
PKCθ releases IL-1β through activation of NFκB pathway in PAs. (**A**) Volcano plot showing the distribution of DEGs of osteogenic cytokines in PRKCQ high expression group versus PRKCQ low expression group in PAs (The data analyzed are described in Figure 2A; OGCs, osteogenic cytokines). (**B**,**C**) Immunohistochemistry for IL-1β in clinical specimens of GH-PAs (*n* = 20) and PRL-PAs (*n* = 20) (**B**) (×200; scale bar, 100 μm). Quantification, right (*** *p* < 0.001; ** *p* < 0.01). The relevance of expression levels of IL-1β and PKCθ (**C**). (**D**) ELISA to detect the secretion level of IL-1β levels of GH3 and ATT20 in treatment with sotrastaurin and PMA (*n* = 6 independent experiments; *** *p* < 0.001). (**E**) ELISA to detect the secretion level of IL-1β levels of GH3 and ATT20 cells in groups of vector and shPKCθ with PMA (*n* = 6 independent experiments; *** *p* < 0.001). (**F**) Schematic diagram of GH3 or ATT20 CM were administered to induce RAW264.7 differentiation towards osteoblasts. (**G**) TRAP staining of RAW264.7 administrated IL-1β recombinant protein and CM of GH3 cells in groups of vector and shPKCθ (×40; scale bar, 500 μm). Quantification, right (*n* = 6 independent experiments; *** *p* < 0.001). (**H**) HE staining exhibited bone destruction of xenograft tumor of GH3 while shPKC and administration of IL-1β recombinant protein (×200; scale bar, 100 μm). Quantification of bone erosion score, right (*n* = 6 mice per group; *** *p* < 0.001). (**I**) TRAP staining of RAW264.7 administered IL-1β mAb and CM of GH3 cells in treatment of PMA (×40; scale bar, 500 μm). Quantification, right (*n* = 6 independent experiments; *** *p* < 0.001). (**J**) HE staining exhibited bone destruction of xenograft tumor of GH3 while administered IL-1β mAb and PMA (×200; scale bar, 100 μm). Quantification of bone erosion score, right (*n* = 6 mice per group; *** *p* < 0.001). (**K**–**N**) WB detection of NF-κB pathway in GH3 (**K**,**M**) and ATT20 (**L**,**N**) while administered PMA combine with sotrastaurin (**K**,**L**) or shPKCθ (**M**,**N**). (**O**) STRING protein–protein interaction network analysis exhibiting proteins interacted with PKCθ. (**P**) CoIP reveals the ability of PKCθ to bind IKKα, P-IKKα/β, IKBα, and p-IKBα while treated with PMA. (**Q**) Schematic diagram exhibited PKC transcriptionally regulates IL-1β through NF-kB pathway.

**Figure 6 cancers-15-01624-f006:**
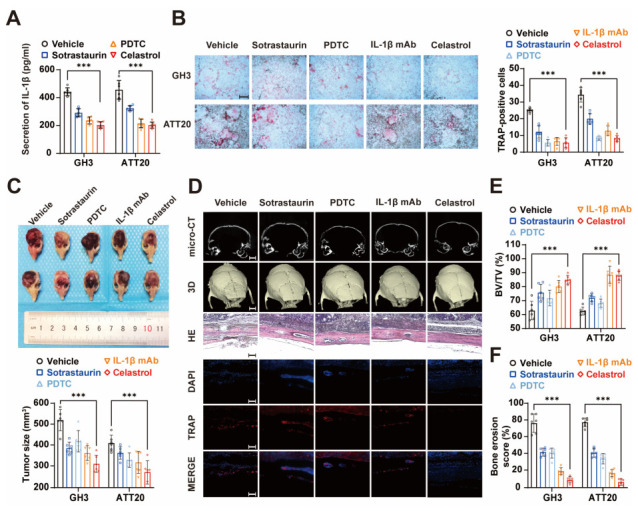
The role of celastrol in reducing bone invasion in PAs. (**A**) ELISA to detect the secretion level of IL-1β levels of GH3 and ATT20 in treatment with sotrastaurin, PDTC, and celastrol (*n* = 6 independent experiments). (**B**) TRAP staining of RAW264.7 were cocultured with GH3 or ATT20 while in treatment with sotrastaurin, PDTC, IL-1β mAb, and celastrol (×40; scale bar, 500 μm). Quantification, right (*n* = 6 independent experiments; *** *p* < 0.001). (**C**) Photographs of GH3 calvaria xenograft tumor specimens in treatment with sotrastaurin, PDTC, IL-1β mAb, and Celastrol. Quantification of tumor size, bottom (*n* = 6 mice per group; *** *p* < 0.001; NS *p* ≥ 0.05). (**D**–**F**) Micro-CT, 3D-reconstruction, HE staining, and immunofluorescence of TRAP+ cells in the tumor–bone junction area (**D**) (×200; scale bar, 100 μm). Quantification of bone volume/tissue volume (BV/TV) (**E**) and bone erosion score (**F**) (*n* = 6 mice per group; *** *p* < 0.001).

**Table 1 cancers-15-01624-t001:** Summary of clinical patient features on BIPAs versus non-BIPAs.

	Level	Overall (*n* = 40)	BIPAs (*n* = 20)	Non-BIPAs (*n* = 20)	*p*-Value
Subtype (%)	GH	20 (50.00)	10 (50.00)	10 (50.00)	1 *
	PRL	20 (50.00)	10 (50.00)	10 (50.00)	
Age (median [IQR])		46.000 [39.000, 53.250]	48.500 [42.000, 55.000]	45.000 [34.000, 48.500]	0.0784 **
Gander (%)	Female	21 (52.50)	11 (55.00)	10 (50.00)	1 *
	Male	19 (47.50)	9 (45.00)	10 (50.00)	
Knosp grade (%)	<Ⅲ	18 (45.00)	4 (20.00)	14 (70.00)	0.0036 ***
	≥Ⅲ	22 (55.00)	16 (80.00)	6 (30.00)	
Diameter (%)	<1 cm	20 (50.00)	5 (25.00)	15 (75.00)	0.0044 *
	≥1 cm	20 (50.00)	15 (75.00)	5 (25.00)	
Ki-67 (%)	<3%	19 (47.50)	5 (25.00)	14 (70.00)	0.0113 *
	≥3%	21 (52.50)	15 (75.00)	6 (30.00)	

* *p*-value of chi-square test. ** *p*-value of Kruskal–Wallis test. *** *p*-value of Fisher’s exact test. Abbreviations: IQR, interquartile range.

## Data Availability

The datasets in this study are available from the corresponding author on reasonable request.

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
