# Peer review of "PKCθ Regulates Pituitary Adenoma Bone Invasion by Activating Osteoclast in NF-κB/IL-1β-Dependent Manner"

_cancers, 2023, doi:10.3390/cancers15051624_

Round 1

Reviewer 1 Report

Authors explored the interaction between PAs tumor cells and osteoclasts, showing that IL-1β drives macrophage-osteoclast differentiation and aggravates the progression of the bone invasion. Moreover PKCθ signaling was established ato promote PAs bone invasion. They found that Celastrol, reduce the secretion of IL-1β and the progression of bone invasion

Please,

1. Spell CT and MRI line 118 and I.P. line 137

2. Line 166 Brüker is the correct name of the company

3. Specify the origin for each cell line

4. Describe in depth how Ki-67 positive cells, wound healing area, invasion rate, and bone erosion score were calculated

5. How were RAW264.7 cells differentiated into OCs?

6. Provide the RIPA and IP buffer recipe

7. What is the significance of IL-10 in figure 1D?

8. TNF-a in fig 1:  it is difficult to see the differences between the two panels

9. Make larger red and blue squares in the legend of figure 2g

10. PMA ATT20 fig 3: why is it not significant?

11. Figures 3b and 3d are too small to read

12. Add "number" in the y-axis label for the graphs in fig 3g and 3h

13. In the WB of fig 5 please show the phospho protein first and below it the total protein

14. The list of supplementary materials is missing

15. Cells were planted, replace with cells were seeded

16. Increase the quality of IF images

17. provide quantification for data obtained by micro-CT

18. Check panels ' letters in fig 5.

Figures are a bit crowded if you manage to lighten them it could be useful to readers

Author Response

Authors explored the interaction between PAs tumor cells and osteoclasts, showing that IL-1β drives macrophage-osteoclast differentiation and aggravates the progression of the bone invasion. Moreover PKCθ signaling was established ato promote PAs bone invasion. They found that Celastrol, reduce the secretion of IL-1β and the progression of bone invasion

Please,

  1. Spell CT and MRI line 118 and I.P. line 137

Response: We have now revised this according to the comment.

  1. Line 166 Brüker is the correct name of the company

Response: We have now revised this according to the comment.

  1. Specify the origin for each cell line

Response: We have now added this in the text.

  1. Describe in depth how Ki-67 positive cells, wound healing area, invasion rate, and bone erosion score were calculated

Response: We have now added this in the text.

  1. How were RAW264.7 cells differentiated into OCs?

Response: RAW264.7 was used to induce osteoclasts. After implantation of RAW264.7 5x10^3 cells/well into 96-well plates, pituitary tumor conditioned medium or drug treatment was given accompanied with a lower level of RANKL (30ng/ml). The medium was changed every 2 days and the end point of induction was reached after 6 days of culture. We added this part in the text.

  1. Provide the RIPA and IP buffer recipe.

Response: We have now added this in the text.

  1. What is the significance of IL-10 in figure 1D?

Response: IL-10 is an anti-inflammatory cytokine. According to our qPCR data, we did not observed increased level of IL-10 in the invasive PA (P value = 0.18) . However, other pro-inflammatory cytokines, such as IL1B and IL6, are notably higher in the invasive PA, suggesting the participation of inflammation signals in PA invasion.

  1. TNF-a in fig 1:  it is difficult to see the differences between the two panels.

Response: We have now revised these figures for better illustration.

  1. Make larger red and blue squares in the legend of figure 2g

Response: We have now revised this figure for better illustration.

  1. PMA ATT20 fig 3: why is it not significant?

Response: We have now performed analysis using ImageJ software instead of manual counting. Also, we have revised and improved the significance labeling of the figures for better understanding. Indeed, PMA induced elevated clone formation capability of ATT20 cells.

  1. Figures 3b and 3d are too small to read.

Response: We have now revised these figures for better illustration.

  1. Add "number" in the y-axis label for the graphs in fig 3g and 3h

Response: We have now added this in the figures.

  1. In the WB of fig 5 please show the phospho protein first and below it the total protein

Response: We have now revised the figures.

  1. The list of supplementary materials is missing

Response: We have now added it.

  1. Cells were planted, replace with cells were seeded

Response: We have now revised this in the text.

  1. Increase the quality of IF images

Response: We have now revised this according to the comment.

  1. provide quantification for data obtained by micro-CT

Response: We have now added it.

  1. Check panels ' letters in fig 5.

Response: We have now revised this according to the comment.

  1. Figures are a bit crowded if you manage to lighten them it could be useful to readers.

Response: We have now re-organized some figures for better illustration.

Reviewer 2 Report

The manuscript by Wang et al. demonstrates a mechanism by which pituitary adenomas (PAs) result in bone invasion through an interaction between Pas tumor cells and osteoclasts.  The authors demonstrate that PAs that result in bone invasion create a more inflammatory environment by releasing proinflammatory cytokines such as TNF-a and IL-1.  The authors went on to note that PKC was more highly expressed in the more aggressive PAs.  Using both chemical inhibitors and sh-RNA against PKC the authors demonstrated that clonal expansion, migration, and proliferation was significantly down regulated in PA tumor cells using multiple cells.  Tumor cells expressing shRNA against PKC produce smaller tumors and less bone erosion compared to tumor control cells. The authors conclude their experiments with several sets of animal and in vitro experiments demonstrating that osteoclasts are involved in the increase in bone erosion.  Osteoclasts are activated by an increase in IL-1b expression which is regulated by an increase in NF-kB activity. 

1.      Authors should explain in material and methods how bone erosion was measured.  From the H and E-stained sections (Figures 3L, 4F, 5H and J, 6F, D and J) it is not clear what is being measured how that measurement is being determined. 

2.     In Figures 4F and 6F in the panels with the TRAP-stained cells it not easy to visualize the TRAP stained cells.  This is also true for the DAPI and F4/80 stained images.  Those panels should be replaced with images where the stained cells can be visualized by the reader.  

3.     In the text the authors write several times “macrophage-osteoclast” differentiation.  While the authors are correct that macrophages and osteoclasts have a common precursor, differentiation of macrophages and osteoclasts are regulated by different signals pathways.  The text should be corrected to make it clearer to the reader that there are different mechanisms that regulate the two cell types.  

4.     In relation to the comment in point #3 it is not clear what the authors are suggesting with the clodronate experiment in Figure 4.  Loss of macrophages with clodronate exposure reduces macrophages but how do the authors propose that affects bone erosion?  Macrophages do not resorb bone.  Clodronate inhibit osteoclast activity, but would the authors expect to see less TRAP positive cells?  Bisphosphonates affect osteoclast activity but not number of osteoclasts.  

5.     Number of animals and number of times that in vitro experiments were performed should be included in figure legends.  

Author Response

The manuscript by Wang et al. demonstrates a mechanism by which pituitary adenomas (PAs) result in bone invasion through an interaction between Pas tumor cells and osteoclasts.  The authors demonstrate that PAs that result in bone invasion create a more inflammatory environment by releasing proinflammatory cytokines such as TNF-a and IL-1.  The authors went on to note that PKC was more highly expressed in the more aggressive PAs.  Using both chemical inhibitors and sh-RNA against PKC the authors demonstrated that clonal expansion, migration, and proliferation was significantly down regulated in PA tumor cells using multiple cells.  Tumor cells expressing shRNA against PKC produce smaller tumors and less bone erosion compared to tumor control cells. The authors conclude their experiments with several sets of animal and in vitro experiments demonstrating that osteoclasts are involved in the increase in bone erosion.  Osteoclasts are activated by an increase in IL-1b expression which is regulated by an increase in NF-kB activity. 

  1. Authors should explain in material and methods how bone erosion was measured.  From the H and E-stained sections (Figures 3L, 4F, 5H and J, 6F, D and J) it is not clear what is being measured how that measurement is being determined. 

Response: We have now added it in the Method & Materials part according to the suggestion.

  1. In Figures 4F and 6F in the panels with the TRAP-stained cells it not easy to visualize the TRAP stained cells.  This is also true for the DAPI and F4/80 stained images.  Those panels should be replaced with images where the stained cells can be visualized by the reader.  

Response: We have now improved the quality of the images according to the comment.

  1. In the text the authors write several times “macrophage-osteoclast” differentiation.  While the authors are correct that macrophages and osteoclasts have a common precursor, differentiation of macrophages and osteoclasts are regulated by different signals pathways.  The text should be corrected to make it clearer to the reader that there are different mechanisms that regulate the two cell types.  

Response: Indeed, the precursor of osteoclasts is generally considered to be monocytes. We have now revised the terms using “macrophage” into “monocyte” when describing osteoclast differentiation as suggested by the reviewer. We thank for the suggestion. 

  1. In relation to the comment in point #3 it is not clear what the authors are suggesting with the clodronate experiment in Figure 4.  Loss of macrophages with clodronate exposure reduces macrophages but how do the authors propose that affects bone erosion?  Macrophages do not resorb bone.  Clodronate inhibit osteoclast activity, but would the authors expect to see less TRAP positive cells?  Bisphosphonates affect osteoclast activity but not number of osteoclasts.  

Response: Clodronate inhibits bone resorption through induction of osteoclast apoptosis according to previous publications[1-3]. Also, Clodronate exhibit monocyte depletion effect[4, 5]. Therefore, the notable depletion of monocytes, as precursor of osteoclast, will result in loss of origins of osteoclast. Taken together, we utilized Clodronate to achieve osteoclast loss and inhibit osteoclast activity. As expected, Clodronate efficiently decreased the number of TRAP positive cells in the bone adjacent region of tumor. We have now revised the text as question #3 for better understanding.

  1. Number of animals and number of times that in vitro experiments were performed should be included in figure legends.  

Response: We have now added the numbers in the figure legends part according to the suggestion.

Reference:

  1. Frediani, B., L. Cavalieri, and G. Cremonesi, Clodronic acid formulations available in Europe and their use in osteoporosis: a review. Clin Drug Investig, 2009. 29(6): p. 359-79.DOI: 10.2165/00044011-200929060-00001.
  2. Vaananen, K., Mechanism of osteoclast mediated bone resorption--rationale for the design of new therapeutics. Adv Drug Deliv Rev, 2005. 57(7): p. 959-71.DOI: 10.1016/j.addr.2004.12.018.
  3. Oelzner, P., R. Brauer, et al., Periarticular bone alterations in chronic antigen-induced arthritis: free and liposome-encapsulated clodronate prevent loss of bone mass in the secondary spongiosa. Clin Immunol, 1999. 90(1): p. 79-88.DOI: 10.1006/clim.1998.4623.
  4. Liu, P., Y. Gao, et al., Glucocorticoid-induced expansion of classical monocytes contributes to bone loss. Exp Mol Med, 2022. 54(6): p. 765-776.DOI: 10.1038/s12276-022-00764-6.
  5. Lehmann, M.L., J.D. Samuels, et al., CCR2 monocytes repair cerebrovascular damage caused by chronic social defeat stress. Brain Behav Immun, 2022. 101: p. 346-358.DOI: 10.1016/j.bbi.2022.01.011.

Round 2

Reviewer 1 Report

The manuscript can be accept in the current form.